# Dramatic Responses to High-Dose Ipilimumab Plus Temozolomide After Progression on Standard- or Low-Dose Ipilimumab in Advanced Melanoma

**DOI:** 10.3390/curroncol32030144

**Published:** 2025-02-28

**Authors:** Julie Williamson, Muhammad Zaki Hidayatullah Fadlullah, Magdalena Kovacsovics-Bankowski, Berit Gibson, Umang Swami, Alyssa Erickson-Wayman, Debra Jamison, Dan Sageser, Joanne Jeter, Tawnya L. Bowles, Donald M. Cannon, Ben Haaland, Joyce D. Schroeder, David A. Nix, Aaron Atkinson, John Hyngstrom, Jordan McPherson, Aik-Choon Tan, Siwen Hu-Lieskovan

**Affiliations:** 1Department of Internal Medicine, School of Medicine, University of Utah, Salt Lake City, UT 84112, USA; julie.williamson@hsc.utah.edu (J.W.);; 2Huntsman Cancer Institute, Salt Lake City, UT 84112, USAberit.gibson@hci.utah.edu (B.G.); daniel.sageser@hci.utah.edu (D.S.); donald.cannon@hci.utah.edu (D.M.C.); aaron.atkinson@utah.edu (A.A.); john.hyngstrom@hci.utah.edu (J.H.);; 3Intermountain Health, Murray, UT 84107, USA; tawnya.bowles@imail.org

**Keywords:** advanced melanoma, refractory melanoma, immune checkpoint inhibitors, translational research, molecular profiling

## Abstract

Patients with advanced melanoma who progress on standard-dose ipilimumab (Ipi) + nivolumab continue to have poor prognosis. Studies support a dose–response activity of Ipi, and one promising combination is Ipi 10 mg/kg (Ipi10) + temozolomide (TMZ). We performed a retrospective cohort analysis of patients with advanced melanoma treated with Ipi10 + TMZ in the immunotherapy refractory/resistant setting (n = 6, all progressed after prior Ipi + nivolumab), using similar patients treated with Ipi3 + TMZ (n = 6) as comparison. Molecular profiling by whole-exome sequencing (WES) and RNA-sequencing (RNA-seq) of tumors harvested through one responder’s treatment was performed. With a median follow up of 119 days, patients treated with Ipi10 + TMZ had a statistically significant longer median progression-free survival of 144.5 days (range 27–219) vs. 44 (26–75) in Ipi 3 mg/kg (Ipi3) + TMZ, *p* = 0.04, and a trend of longer median overall survival of 154.5 days (27–537) vs. 89.5 (26–548). Two patients in the Ipi10 + TMZ cohort had a partial response, and both responders had BRAF V600E mutant melanoma. RNA-seq showed enrichment of inflammatory signatures, including interferon responses in metastases after Ipi10 + TMZ compared to the primary tumor, and downregulated negative immune regulators. Ipi10 + TMZ demonstrated efficacy, including dramatic responses in patients refractory to prior Ipi + anti-PD1. Molecular data suggest a potential threshold of Ipi dose for activation of sufficient anti-tumor immune response, and higher doses are required for some patients.

## 1. Introduction

The advent of checkpoint inhibitor-based immunotherapy has drastically changed the treatment and prognosis of advanced melanoma in the past decade. The combination of ipilimumab (Ipi) 3 mg/kg plus nivolumab 1 mg/kg has become the standard of care, achieving 52% overall survival and 36% progression-free survival at 5 years [1]. However, this leaves an unmet need for the majority of patients who still progress and those who do not survive past 5 years. New immunotherapy agents are entering the market, with relatlimab, an antibody against LAG-3, the latest to gain FDA approval for advanced melanoma. However, the complete response rate to relatlimab plus nivolumab in untreated advanced melanoma was 16.3% and the partial response rate was 26.8%, which unfortunately still leaves many with progression of their disease [2,3]. Patients with symptomatic brain metastases continue to fare poorly, with 36% overall survival and 19% overall survival at 3 years on a combination of Ipi and nivolumab [4]. Dosing of this immunotherapy combination has been studied, although primarily within the lens of safety profiles. Flipped dosing with Ipi 1 mg/kg plus nivolumab 3 mg/kg had significantly reduced rates of severe adverse effects while maintaining similar progression-free and overall survival at 3 years, so higher doses are not routinely prescribed or studied [5]. However, before combination therapy became the standard of care, an Italian study demonstrated a significant improvement in overall survival with Ipi 10 mg/kg compared to 3 mg/kg [6]. Additionally, using standard-dose Ipi following progression on low-dose Ipi has shown clinical activity in patients with metastatic melanoma [7]. This evidence supports the dose–response activity of Ipi.

One promising combination for these patients is Ipi 10 mg/kg plus temozolomide (TMZ). TMZ works by depleting regulatory T cells and suppressing their function, and it may enhance the antitumor activity of Ipi when dosed together [8,9]. One study found a response rate of 31% using this combination in immunotherapy naïve patients with advanced melanoma, whereas Ipi 3 mg/kg and 10 mg/kg alone had response rates of 12% and 15%, respectively [6,10,11]. TMZ crosses the blood–brain barrier and therefore may improve treatment efficacy in patients with brain metastases.

The aim of this retrospective case series is to describe outcomes for patients with advanced melanoma who were treated with high-dose (10 mg/kg) Ipi plus TMZ in the immunotherapy refractory/resistant setting, using a cohort of similar patients treated with standard-dose (3 mg/kg) Ipi plus TMZ as comparison. Two patients with central nervous system (CNS) involvement demonstrated an extraordinary response to Ipi 10 mg/kg plus TMZ, despite progressing on regular- or low-dose IPI plus nivolumab and other treatments previously.

## 2. Materials and Methods

A retrospective chart review was performed to identify patients with advanced melanoma treated with Ipi plus TMZ at Huntsman Cancer Institute from 1 June 2014 to 30 June 2022, under University of Utah approved institutional review boards (IRBs) # 00138167. Epic (Epic System Corporation) Workbench Reporting was used to determine all patients who received a Beacon treatment plan containing the combination of IPI and TMZ during the above study time horizon. Epic is a healthcare software company based in the United States that provides electronic health system records (EHRs) for managing patient records. Patients were excluded if they did not have advanced melanoma; therefore, one patient with advanced Merkel cell carcinoma was identified and excluded. Two cohorts were identified; cohort A (n = 6 patients) included those treated with high-dose Ipi (10 mg/kg) plus TMZ, and cohort B (n = 6 patients) included those treated with standard-dose Ipi (3 mg/kg) plus TMZ. Ipi was dosed on day 1 every 3 weeks for 4 cycles. TMZ was dosed as 200 mg/m^2^ on days 1–4 every 3 weeks for 4 cycles.

### 2.1. Sample Collection and Preparation

An archival specimen collected from one patient included a punch biopsy of the primary site, an ultrasound-guided core needle biopsy of the liver, and a core needle biopsy of a lesion on the back. All samples were placed in formalin and made into Formalin-Fixed Paraffin-Embedded (FFPE) blocks. The samples were stained with Melan-A (MilliporeSigma, St Louis, MO, USA/Clone A103/281M-8), and SOX10 (Millipore-Sigma, St Louis, MO, USA/Clone EP268/AC-0237) and had clinical pathology review to confirm diagnosis. Unstained slides were submitted to Tempus (Chicago, IL, USA) for sequencing.

### 2.2. Whole-Exome Sequencing

Whole-exome sequencing of the samples was performed at the Tempus Lab, Inc. (Chicago, IL, USA) with the Tempus xE assay. Fastq were aligned to the human reference genome GRCh38 using BWA [12]. Next, polymerase chain reaction (PCR) duplicates were removed, and somatic variants—Insertion–Deletion Variants (INDELs) and Single-Nucleotide Variants (SNVs)—were called using Illumina’s Manta and Strelka2 software (https://github.com/HuntsmanCancerInstitute/Workflows) [13]. The alignment, variant calling, and variant annotations were integrated in containerized Snakemake workflows that are publicly available at https://github.com/HuntsmanCancerInstitute/Workflows. Next, the mutations were filtered to retain deleterious mutations; that is, if a mutation falls into protein coding genes and is considered ‘deleterious’ according to Sorting Intolerant From Tolerant (SIFT) [14] and ‘probably damaging’ or ‘possibly damaging’ according to Polymorphism Phenotyping version 2 (PolyPhen-2) [15]. Variant annotations were carried out using the Ensembl Variant Effect Predictor tool [16]. The deleterious mutations list is included as Supplemental Appendix A. Tumor Mutational Burden (TMB) calculation was based on the number of deleterious mutations divided by the length of human exons (38 Mb). A phylogenetic tree was constructed based on the Neighbor joining method implemented in the R phangorn package (version 2.12.1) [17]. Pathway enrichment analysis was based on the web-based tool Enrichr [18], using the Hallmark 2020 pathway [19].

### 2.3. RNA-Sequencing

Samples were sequenced by the Tempus Lab, Inc. (Chicago, IL, USA). RNA-seq fastq was aligned with STAR aligner [20] and gene expression was quantified using RSEM (https://github.com/HuntsmanCancerInstitute/Workflows) [21]. To determine enriched gene signatures in the samples, Single-Sample Gene Set Enrichment Analysis (ssGSEA) was conducted using the curated Hallmark gene sets (h.all.v2022.1.Hs.symbols.gmt). The R package Gene Set Variation Analysis (GSVA) (version 2.0.1) [22] was used to calculate the normalized ssGSEA pathway score and the normalized results were scaled and displayed as a heatmap. Estimation of Stromal and Immune cells in Malignant Tumours using Expression data (ESTIMATE) [23] analysis (version 1.0.13) was performed to infer the presence of stromal and immune cells in the tumor using gene expression data. All RNA-seq analysis was carried out using the RSEM-provided transcript-per-million (TPM) values.

### 2.4. Statistical Analysis

All statistical analyses were conducted using R version 4.4.1. Patients were stratified into two groups based on ipilimumab dosing: high-dose ipilimumab (10 mg/kg, denoted as “Ipi10”) and low-dose ipilimumab (3 mg/kg, denoted as “Ipi3”). Comparisons between these two groups were performed using a two-sided Wilcoxon rank sum test (implemented using wilcox.test function). A *p*-value of less than 0.05 was considered statistically significant. To visualize the data distribution, 95% confidence intervals for the group means were calculated using bootstrapping methods. Bootstrapping was employed, given the small sample size and assumption of a potential deviation from normality. Bootstrapping was performed using the R package boot (version 1.3-31) with 1000 random samples with replacement. The bootstrapping confidence interval highlights 95% of the (between the 2.5th and 97.5th percentile) values.

Survival analyses were performed to compare progression-free survival (PFS) and overall survival (OS) between the Ipi10 and Ipi3 group. OS was defined from the time of ipilimumab therapy to date of death. PFS was defined as the time of ipilimumab therapy to the date of documented disease progression. For PFS, disease progression was categorized as either progressive disease (PD) or mixed response (MR) according to the response evaluation criteria in solid tumors (RECIST). The survival package (version 3.7-0) was utilized for survival analyses, and the survminer package (version 0.4.9) was used to generate Kaplan–Meier plots with median survival times. The log-rank test was used to determine significance in the Kaplan–Meier analysis. A *p*-value of less than 0.05 was considered statistically significant.

## 3. Results

### 3.1. Case Study 1

A 31-year-old female (patient 10A) was initially diagnosed with stage IIIc BRAF V600E positive nodular melanoma on the skin of the buttock. She was started on neoadjuvant pembrolizumab 200 mg IV every 3 weeks in a clinical trial setting. After three cycles, she was found to have stage IV disease progression (pT4b, N3c, M1c), with lymph node and liver metastases by the time of resection. She was then started on dabrafenib, trametinib, and pembrolizumab for four months. Pembrolizumab was intermittently held due to toxicity. She initially had a partial response but then had progression of disease in her liver, lungs, and lymph nodes. She was started on flipped-dose Ipi (1 mg/kg) plus nivolumab (3 mg/kg) due to previous toxicity with pembrolizumab and tolerated three cycles. She had a mixed response in her liver, stable disease in the lung and lymph nodes, but had progression to her bones and brain with new spine and femur lesions and two new 9 mm lesions in the left frontal and left occipital lobes. Her therapy was ultimately switched to high-dose Ipi (10 mg/kg) plus TMZ, and she received stereotactic radiosurgery to the brain lesions. Her subsequent scans showed decreased brain lesions and, ultimately, she had complete resolution of her brain metastases and no new lesions by 3 months of treatment. After four cycles, she developed grade 3 hepatitis and was treated with high-dose steroids with an extended period of steroid taper. While further treatment was held, she developed relapsed disease with new spinal, pulmonary, and liver lesions 5 months from the last dose of Ipi. Her lactate dehydrogenase (LDH) also peaked at this time to 2294. She was treated with standard-dose Ipi (3 mg/kg) plus nivolumab for four cycles followed by maintenance nivolumab and had another dramatic partial response (Figure 1) which lasted (at present, no progression at 10 months of follow up). At 17 months since high-dose Ipi was initiated, magnetic resonance imaging (MRI) of the brain still revealed no evidence of CNS disease. Whole-body computed tomography (CT) scan showed continued a partial response except for one left-ovarian mass that increased in size from 4.6 cm to 5.4 cm. She was taken to the operating room and underwent surgical metastatectomy of this ovarian mass. Pathology analysis revealed extensive tumor necrosis and histiocytic inflammation but no viable tumor was identified.

To further understand patient 10A’s treatment response, we analyzed molecular profiling by RNA-seq and whole-exome sequencing (WES) of the tumors harvested throughout her treatment, including the surgical specimen at the initial site of diagnosis (P1, harvested from skin of the buttock) and subsequent site of metastasis to the liver (M2) 2 days after the first dose of low-dose Ipi plus nivolumab and prior to the Ipi10 plus TMZ treatment, and to the subcutaneous soft tissue in the back (M3), after disease progression to Ipi10 plus TMZ and prior to subsequent Ipi3 plus nivolumab (Figure 1A and Figure 2A). WES revealed only 12 shared somatic mutations among these tumors (Figure 2B), including the BRAF V600E mutation, and other genes important in defining melanoma and continued activation of the MAPK pathway. Calculations of tumor mutational burden (TMB) indicate the liver metastasis (M2) has higher TMB 7.5 mutations/Mb than the primary tumor (P1), with 1.4 mutations/Mb, and the later soft tissue metastasis (M3), with 2.7 mutations/Mb. We focused on the genes mutated in the immune- and inflammation-related pathways from the Hallmark gene sets and found that the interferon alpha and gamma response pathways were enriched in the genes exclusively mutated in the M2 sample (Figure 2C). Interestingly, IFNGR2 is exclusively mutated in M3, which may play a role in altering the response to immunotherapy. To better understand the transcriptome changes during treatment, single-sample gene set enrichment analysis (ssGSEA) and ESTIMATE analysis was performed on the RNA-seq of these tumors. ESTIMATE analysis revealed that the Liver metastatic (M2) has the highest Immune score. This observation was also supported by ssGSEA, where M2 showed an enrichment of immune-related gene sets including interferon alpha and gamma responses (Figure 2D). M3, when compared to P1, is also enriched with the inflammatory response gene set. On the other hand, pathways shown to be negative regulators of anti-tumor immune response, such as Wnt signaling and TGF beta signaling, are downregulated in both M2 and M3. RNA-Seq also revealed upregulation of alternative signaling pathways such as PI3K/AKT, Notch, and Hedgehog signaling in M3 when compared to P1.

### 3.2. Case Study 2

A 52-year-old male (patient 10B) also responded to high-dose Ipi plus TMZ after progressing on multiple lines of therapy. At age 45, he was initially diagnosed with stage IIIC (pT3bN3M0) superficial spreading melanoma to the right forehead. He underwent surgical resection and sentinel lymph node biopsy, as well as right parotidectomy (due to positive sentinel lymph node involvement) and participated in an adjuvant vaccine trial. Three years later, he developed a single-lung metastasis to the left lower lobe. He underwent wedge resection of the lung where this lesion was involved and was started on adjuvant nivolumab. After 5 months of treatment (doses were intermittently held due to toxicity), he developed metastases to the left-sided pleura. Next-generation sequencing revealed a BRAF V600E mutation, so triple therapy with dabrafenib, trametinib, and pembrolizumab was started. He initially had a partial response to this regimen but 16 months later his melanoma progressed and metastasized to the brain. This was initially managed by SRS to the brain lesion and continued triple therapy, but systemic disease eventually progressed by 7 months. He started Ipi (initially 1 mg/kg but increased to 3 mg/kg cycles 2–4) plus nivolumab and tolerated all four cycles of combination and two more doses of monotherapy nivolumab. However, his disease continued to worsen in his CNS, with three new lesions, and he was started on encorafenib and binimetinib. His disease initially stabilized, then progressed both systemically and in the CNS within 10 months. High-dose Ipi (10 mg/kg) plus TMZ were started. After three cycles, his positron emission tomography (PET) CT revealed partial response and near complete resolution of fluorodeoxyglucose (FDG) activity of his disease burden (Figure 3).

### 3.3. Cohort

Due to the dramatic response to high-dose Ipi plus TMZ in the cases detailed above, we were interested in analyzing the outcomes of other patients with advanced melanoma treated with this combination at our institution. Of the 12 patients with advanced melanoma treated with Ipi plus TMZ, 6 were treated with high-dose Ipi (10 mg/kg), and 6 were treated with standard-dose Ipi (3 mg/kg) (Table 1). Age and BRAF mutations were similar between both groups. All but one patient was previously treated with Ipi. Notable differences between groups include gender as well as prior exposure to Ipi plus nivolumab. All patients treated with high dose Ipi had prior treatment with Ipi plus nivolumab, whereas only half of patients treated with standard-dose Ipi did (Table 1).

This is a very heavily pre-treated patient population, with most patients only receiving one dose before moving on to hospice or passing away from disease progression (patients 10E, 10F, 3C, 3D, 3E, 3F). All patients treated with regular-dose Ipi plus TMZ had progression of disease as their best response, while two patients treated with high-dose Ipi plus TMZ had partial response at the first response evaluation (described in the case section), and a third patient had a mixed response.

At a median follow up of 119 days, patients treated with high-dose Ipi plus TMZ had statistically significant longer median progression-free survival (PFS) of 144.5 days (27–219) vs. 44 days (26–75) for standard-dose Ipi and TMZ, *p* = 0.04 (Figure 4). Due to the small sample size in each cohort and heterogeneous prior treatment history, no statistically significant differences can be drawn from subgroup analysis; however, there is also a similar trend of longer median PFS for patients who had more than one cycle of treatment, 144.5 days (96–219) for high-dose Ipi plus TMZ vs. 61 days (47–75) for standard-dose Ipi plus TMZ, and in patients who were previously exposed to regular-dose Ipi plus nivolumab, 144.5 days (27–219) vs. 39 days (26–55), respectively (Figure 4). There is no statistical difference in terms of median overall survival (OS) but a trend of longer median OS of 154.5 days (27–537 days) vs. 89.5 days (26–548 days) for high- and standard-dose Ipi plus TMZ. This trend is more evident in patients who were previously exposed to regular-dose Ipi plus nivolumab; median OS in the high-dose Ipi group was 154.5 days (27–537) vs. 39 days (26–55) for the standard-dose Ipi group (Figure 4).

## 4. Discussion

This cohort and these case reports highlight the potential for high-dose Ipi (10 m/kg) plus TMZ in advanced melanoma, including in patients who are heavily pre-treated or have CNS involvement. This study represents a small cohort of patients and is not a randomized controlled trial. However, patients who have progressed on several immunotherapy regimens with poor performance status are often precluded from any clinical trials, and different combinations or dosages are often attempted as a last resort option. Both patients who had a partial response in this report fell into this category and both patients demonstrated remarkable response to Ipi 10 mg/kg plus TMZ, with near complete response of systemic disease and complete remission of CNS disease, despite progressing on all previous regimens including combination Ipi and nivolumab as well as combination immunotherapy with targeted therapy. This confirms a dose–response correlation of Ipi dosing. High-dose Ipi is a reasonable regimen for patients who have already progressed on a prior lower dose of Ipi, and the toxicity profile can be successfully managed.

Dosing of anti-PD1 inhibitors does not seem to change efficacy, because these antibodies work by binding to receptors expressed on activated T cells, and greater concentrations do not alter activity [24,25]. However, the CTLA-4 blockade by Ipi is more nuanced. Previous studies have noted the dose–response relationship of Ipi in advanced melanoma [6,26,27]. One study used escalating doses of Ipi on CD4+ T cells of cancer patients. At lower doses, CTLA4 blockade expanded regulatory T cells. Only at the highest dose studied (3 mg/kg) did Ipi expand effector T cells [28]. Furthermore, inhibiting CTLA4 may allow for the formation of de novo immune responses in addition to removing the inhibition of active effector responses [29]. The mechanism and effects Ipi has on T cell activity appear to be influenced by the dose. After progressing on prior Ipi and nivolumab, patients may still develop a robust and durable response to higher-dose Ipi. This is highlighted in the cohort, where patients who were previously exposed to low- or regular-dose Ipi plus nivolumab responded well to high-dose Ipi plus TMZ but had no response to standard-dose Ipi plus TMZ, suggesting TMZ is not the determinant of these responses. From this cohort, there is no clear evidence that TMZ has any direct effect in this regimen, although it is reasonable to add TMZ for CNS disease, and more studies are needed to elucidate its effect.

Our molecular investigation by WES and RNA-seq of the tumors in patient 10A provided further insight into the potential mechanism of her response to high-dose Ipi10 plus TMZ (Figure 2). The patient’s cutaneous primary tumor P1 represents a “cold tumor”, with very low activity of the interferon alpha and gamma responses and inflammatory gene sets, and highly active TGF beta and Wnt signaling pathways. This status was completely reversed in M2, a liver metastasis sampled 2 days after the first cycle of low-dose Ipi plus nivolumab treatment. There are elevated Immune scores, indicating a robust immune presence within the tumor microenvironment, enriched gene expression and mutations of interferon alpha and gamma response pathways exclusively in the M2 sample, suggesting an enrichment of interferon response and representing a switch to a “hot tumor” after receiving low-dose Ipi plus nivolumab treatment. Clinically, the patient did have a mixed response in the liver, and this sampled liver lesion responded to low-dose Ipi plus nivolumab. However, even after three cycles of low-dose Ipi plus nivolumab, despite continued response in the liver, her disease progressed to the bones and brain, with new spine and femur lesions and two new lesions in the left frontal and left occipital lobes (Figure 1). Remarkably, these lesions later responded to high-dose Ipi plus TMZ, suggesting a potential “threshold” of Ipi dose for the activation of sufficient anti-tumor immune response to control all tumor growth. This threshold could differ in individual patients. For patient 10A, Ipi10 appears to be required for immune control of all her lesions. This provides rationale for a higher dose of Ipi after progression on regular or low doses of Ipi-containing regimen. One alternative explanation of her experience is a delayed response to her previous low-dose Ipi plus nivolumab, and with one case, it is hard to prove either way. However, the experience with patient 10B (another partial responder), whose disease progressed after four cycles of regular-dose Ipi plus nivolumab and two more doses of maintenance nivolumab, yet responded to high-dose Ipi10 plus TMZ (Figure 3), as well as well as for those with significantly prolonged PFS in the high-dose Ipi cohort (Table 1, Figure 4) who all received prior combination of regular- or low-dose Ipi plus nivolumab, support the argument that this is not simply a delayed response to prior Ipi plus nivolumab combination, and warrants further investigation.

To our surprise, WES revealed only 12 shared somatic mutations among these tumors (Figure 2B). The BRAF V600E mutation is present in all samples, suggesting a common clonal origin; however, RNA-Seq revealed upregulation of alternative signaling pathways such as PI3K/AKT, Notch, and Hedgehog signaling in M3 when compared to P1, suggesting potential mechanisms of resistance to interim targeted therapies. The M2 liver metastasis has many more mutations as compared to the primary tumor, possibly due to a post-treatment effect, as it was taken 2 days after low-dose Ipi plus nivolumab was given. Additionally, potential clonal evolution for adaptation to the anatomical liver site may have taken place. M3 is a soft tissue metastasis in the back sampled at the point of disease progression after high-dose Ipi plus TMZ (after a 4-month duration of no treatment and high-dose steroids for liver toxicity) and at baseline prior to regular Ipi plus nivolumab. M3 has 50 unique mutations when compared to P1, and is enriched with the inflammatory response gene set [30], which may explain why the patient responded to regular Ipi plus nivolumab later.

Anti-PD1 agents were not required for the induction of the response; however, they appear to be important after a response to high-dose Ipi to give a durable response. This patient (10A) progressed after high-dose Ipi and TMZ were stopped, but responded again to standard-dose Ipi plus nivolumab and had a lasting response while on maintenance nivolumab only (Figure 1). Since CTLA4 inhibitors may allow for the formation of de novo immune responses, one possible explanation for this outcome is that higher doses may create new anti-tumor responses. Once established, anti-PD1 agents may work through their primary mechanism of unleashing already-established immune responses [29].

Furthermore, these cases highlight the potential for high-dose Ipi and TMZ in brain metastases. This study’s small sample size and retrospective nature limit the generalizability of our findings. We recognize the importance of investigating the efficacy and safety of high-dose ipilimumab plus temozolomide in larger, more diverse patient cohorts. Future prospective studies with expanded patient populations are warranted to validate our results and further elucidate the potential role of this combination therapy in advanced melanoma treatment. A previous study demonstrated a durable response when using this combination in patients with advanced melanoma naïve to immunotherapy, with median overall survival of 24.5 months. However, only 3% of this population had brain metastases [10]. TMZ has been used in treatment of both primary CNS tumors and metastases within the CNS due to its ability to cross the blood–brain barrier [31]. The short dosing interval in this regimen (5 days around the Ipi dosing) is mainly employed to deplete immune suppressive Tregs; however, it is possible that it also exerts direct anti-tumor activity towards the brain lesions. One study of 64 patients with advanced melanoma found a 31% response rate to Ipi 10 + TMZ; however, only 2 of the patients had brain metastases [10]. Given the poor outcomes in patients with melanoma brain metastases, especially after progression on a regular dose of Ipi plus nivolumab, and the dramatic response highlighted in the case report here, further studies are needed to investigate the potential for high-dose Ipi and TMZ in this specific population, as well as in patients with advanced melanoma who have progressed on immune checkpoint inhibitors.

## Figures and Tables

**Figure 1 curroncol-32-00144-f001:**
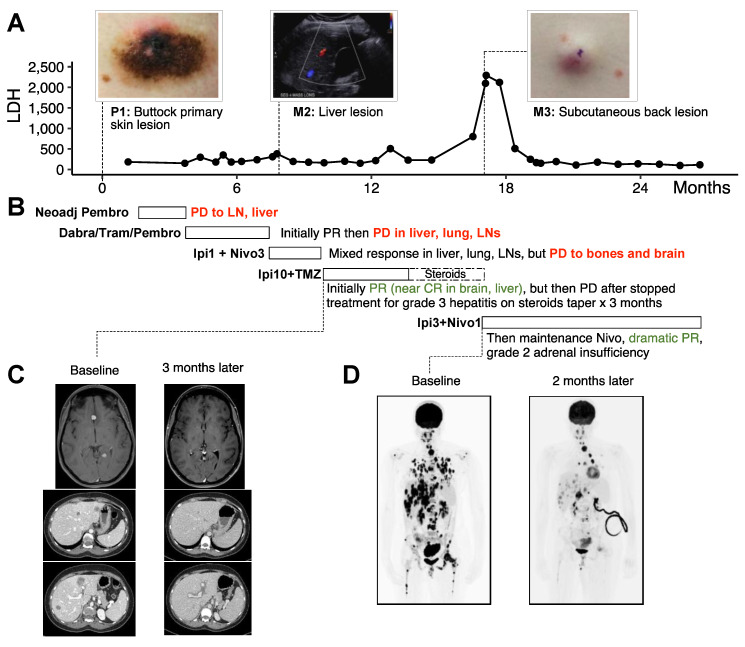
Timeline of patient 10A’s treatment. (**A**) Lactate Dehydrogenase (LDH) is plotted against time. Point of time when specimens were collected for next-generation sequencing profiling is indicated as primary (P) and metastatic (M). (**B**) Timeline describing treatment regime. (**C**) Brain MRI revealing two metastatic lesions prior to first dose of Ipi10 plus TMZ, one at interhemispheric fissure between inferior frontal lobe and one at left paramedian occipital lobe; this was first evidence of CNS involvement. Brain MRI 12.5 weeks later, after four cycles of Ipi 10 mg/kg plus TMZ, revealed complete resolution of anterior nodule and posttreatment changes and completely resolved perilesional vasogenic edema of occipital nodule. (**D**) PET scan 4 months after initiating high-dose steroids and holding treatment for liver toxicity from Ipi10 plus TMZ showed diffuse disease progression. PET scan 9.5 weeks later, after receiving 3 cycles of standard-dose Ipi plus nivolumab showed dramatic partial response. Neoadj: neoadjuvant; pembro: pembrolizumab; dabra: dabrafenib; tram: trametinib; Ipi: ipilimumab; nivo: nivolumab; TMZ: temozolomide; PD: progressive disease; PR: partial response; CR: complete response; LN: lymph node.

**Figure 2 curroncol-32-00144-f002:**
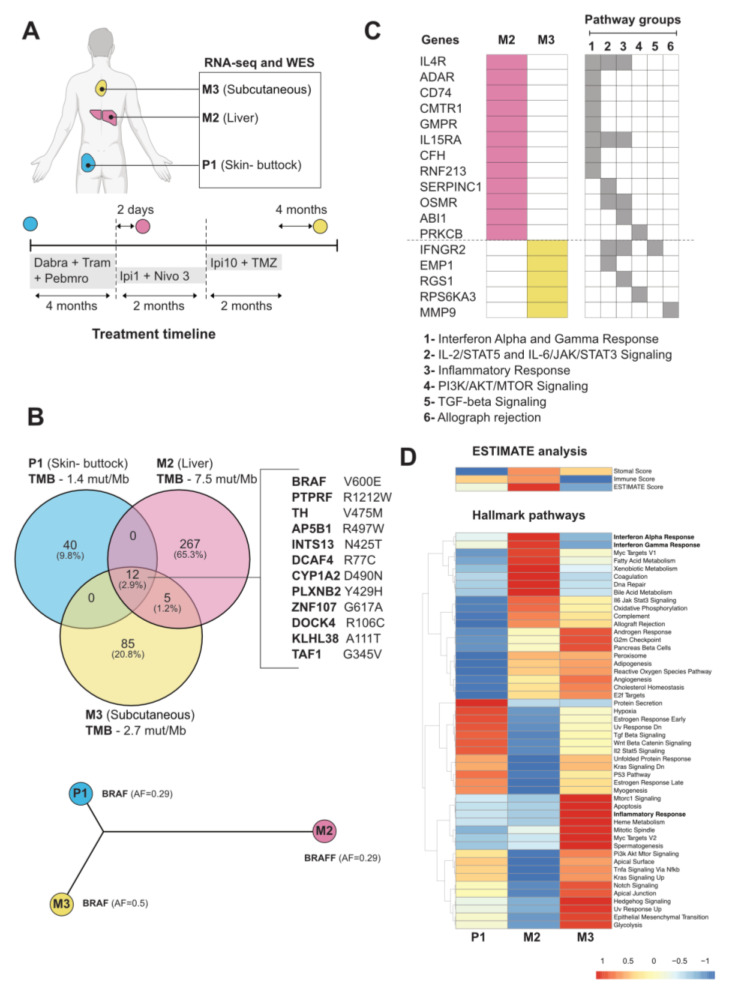
Exome and RNA-seq sequencing of tumors. (**A**) Tumor samples collected at different anatomical sites from patient, labeled P (primary) and M (metastasis). (**B**) Top: overlap of somatic mutations found in tumor, with tumor mutational burden (TMB) value (number of mutations/Mb) presented under each sample. Bottom: construction of phylogenetic tree based on somatic mutations considered deleterious and damaging. Allele frequency (AF) of BRAF V600E is depicted. (**C**) Enriched pathways of genes uniquely mutated in samples M2 and M3. (**D**) Heatmap of ESTIMATE analysis (top) and ‘Hallmark Pathways’ (bottom) showing normalized result of single-sample gene set enrichment analysis.

**Figure 3 curroncol-32-00144-f003:**
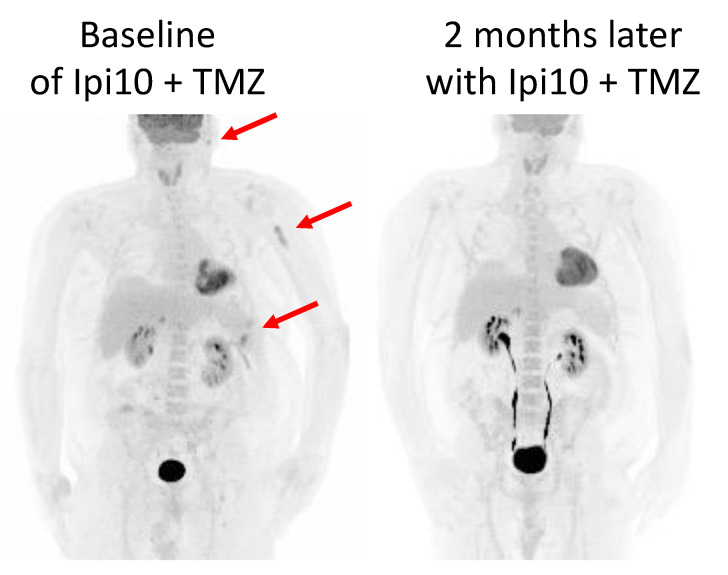
PET CT of patient 10B. PET scan 3 weeks prior to initiation of Ipi10 plus TMZ showed disease progressing with left preauricular nodule, splenic lesions, left proximal humeral bone marrow space, mesenteric nodes, and right-sided level 3 nodes. PET scan 3 months later, after receiving 3 cycles Ipi10 plus TMZ, showed dramatic response with complete resolution of lesions in spleen, mesenteric nodes, left preauricular soft tissue, and left humeral shaft.

**Figure 4 curroncol-32-00144-f004:**
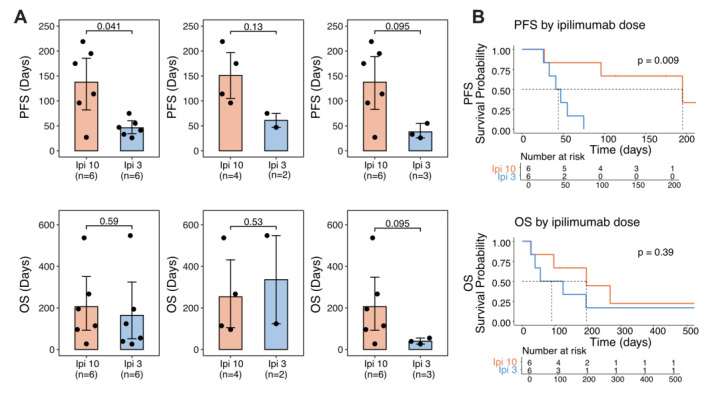
Progression-free survival (PFS) and overall survival (OS) in patients treated with Ipi 10 mg/kg plus TMZ (Ipi 10) vs. Ipi 3 mg/kg plus TMZ (Ipi 3). (**A**) PFS and OS in all patients (left panel), and patients who had >1 cycle of Ipi 10 mg/kg plus TMZ (middle panel), and who had previous exposure to regular- or low-dose Ipi plus nivolumab (right panel). Two-sided Wilcoxon rank sum test was used to determine statistical significance. (**B**) Association of ipilimumab dosage with PFS (top) and OS (bottom). Log-rank test was used to determine statistical significance.

**Table 1 curroncol-32-00144-t001:** Baseline characteristics and outcomes of patients treated with Ipi 10 mg/kg (patients 10A–10F) + TMZ and Ipi 3 mg/kg + TMZ (patients 3A–3F).

Pt ID	Gender	Age at C1D1	Ipilimumab mg/kg	Temodar 200 mg/m^2^ D1–D4	Melanoma Subtype	BRAF	Prior Therapies	No of Cycles	Best Response	PFS	OS	Alive	Next Lines of Therapy	irAE	Grade	HD Steroids?
**10A**	F	30	10	Y	Nodular	V600E	neoadj pembro, D/T/Pembro, **Ipi1+nivo3**	4	**PR**	219	537	Y	ipi3/nivo1	Hepatitis	3	Y
**10B**	M	52	10	Y	Superficial Spreading	V600E	Adj nivo, D/T/pembro, **Ipi3/Nivo1**, E/B	4	**PR**	114	114	Y	None (Continued response)	Rash	3	Y
**10C**	F	48	10	Y	Acral	WT	**Ipi3+nivo1**, Palbociclib+nivo	5	**Mixed response**	175	267	N	lenvatinib + pembro	Diarrhea	2	Y
**10D**	M	35	10	Y	Non-Acral Cutaneous	V600E	Ipi3, Pembro, **Ipi3+Nivo1**, D/T/P	3	PD	96	96	N	None	PE, SOB	3	Y (for brain mets)
**10E**	F	67	10	Y	Acral	WT	Neoadj Nivo, **Ipi3+nivo1**, microbiome+nivo, lenvatinib+pembro	1	PD	195	195	N	None	Diarrhea	3	Y
**10F**	M	66	10	Y	Nodular	V600E	Adj nivo, D/T, E/B, **Ipi3/Nivo1**	1	PD	27	27	N	None	No	0	N
**3A**	M	47	3	Y	Non-Acral Cutaneous	V600E	**Ipi3**, IL-2, Dabra, pembro, D/T	3	PD	75	124	N	Carbo	No	0	N
**3B**	M	54	3	Y	Nodular	V600K	**Ipi3**, D/T, neoadj TVEC, adj nivo, D/T/P	2	PD	47	548	Y	Carbo/Taxol, lenvatinib+pembro, rela+nivo	Eye muscle inflammation	3	Y
**3C**	M	69	3	Y	Acral	WT	TVEC+pembro, Ipi3+Cavatak, **Ipi3+Nivo1**	1	PD	26	26	N	None	No	0	N
**3D**	M	45	3	Y	Non-Acral Cutaneous	V600E	**Ipi3+nivo1**, nivo, D/T, pembro, D/T/P	1	PD	55	55	N	None	No	0	N
**3E**	F	21	3	Y	Nodular	V600E	Adj D/T, Neoadj pembro, **Ipi3+Nivo1**, TVEC+Pembro, temodar 75mg/kg daily	1	PD	39	39	N	None	No	0	N
**3F**	M	52	3	Y	Choroidal	WT	Tram+AKT inh	1	PD	41	194	N	Pembro, Disulfiram/Zinc	No	0	N

## Data Availability

No publicly available data sets were used; however, the workflows used were publicly available and are detailed in the methods section. The alignment, variant calling, and variant annotations were integrated in containerized Snakemake workflows that are publicly available at https://github.com/HuntsmanCancerInstitute/Workflows accessed on 29 October 2024.

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
