# Peer review of "Dramatic Responses to High-Dose Ipilimumab Plus Temozolomide After Progression on Standard- or Low-Dose Ipilimumab in Advanced Melanoma"

_curroncol, 2025, doi:10.3390/curroncol32030144_

Round 1
Reviewer 1 Report
Comments and Suggestions for Authors
I went through the manuscript titled “Dramatic responses to high dose ipilimumab plus temozolomide after progression on standard or low dose ipilimumab in advanced melanoma”. In my view it is an important manuscript which describes a therapeutic intervention (ipilimumab plus temozolomide in two different doses) for melanoma patients that do not respond to standard ipilimumab therapy.
Although the results seem to have substantial importance, the study is limited by its small sample size and retrospective nature. These two points in my view should be better explained in the materials and methods part. Moreover, given these limitations the statistical analysis must be very robust to support any outcome, as the study already has significant limitations. But in the present manuscript the statistical analysis is not reported.
In brief:
Lines 68 – 75 Report 1. The selection criteria of the cases (describe that it was about patients that did not respond to standard of care) 2. The inclusion and exclusion criteria in the two cohorts 3. If other cases were excluded why was that done? 4. Specify EPIC for readers not familiar with the USA health system
Lines 77 – 81 Specify abbreviations and cite in parenthesis the source of the staining reagents. Specify Tempus data in parenthesis.
Lines 83 – 106 Cite all sources when you report brand names
Line 107 The materials and methods part, lack a data analysis part explaining the statistical analysis performed.
Lines 108 – 212 It is not understandable whether these two case studies are included or excluded from the cohorts. In any case it is to be reconsidered how they should be presented. (if excluded must be included, if included they may be presented within each cohort presentation). Moreover, this part should be improved. The writing shall be more scientific- writing like than conference – presentation like.
Lines 214 – 255 There is not evident what the authors mean reporting statistically significant or not significant. Shouldn’t they perform survival analysis? A Kaplan – Mayer chart must be presented also. These lines also must undergo extensive rewrite.
The discussion also must be more precise and refer to data achieved. In my view the discussion should also be rewritten.
Line 364 Report if the consent was written.
Author Response
Reviewer #1
I went through the manuscript titled “Dramatic responses to high dose ipilimumab plus temozolomide after progression on standard or low dose ipilimumab in advanced melanoma”. In my view it is an important manuscript which describes a therapeutic intervention (ipilimumab plus temozolomide in two different doses) for melanoma patients that do not respond to standard ipilimumab therapy.
Although the results seem to have substantial importance, the study is limited by its small sample size and retrospective nature. These two points in my view should be better explained in the materials and methods part. Moreover, given these limitations the statistical analysis must be very robust to support any outcome, as the study already has significant limitations. But in the present manuscript the statistical analysis is not reported.
We greatly appreciate that the reviewer found merit in our manuscript, noting that “it is an important manuscript which describes a therapeutic intervention … for melanoma patients that do not respond to standard ipilimumab therapy.”
We also agree with the reviewer that this study is limited by its small sample size and the retrospective nature should be explained in the material and methods. We have revised the Materials and Methods section to include a detailed description of the patient cohort. Additionally, we included a statistical analysis section within the Material and Methods to outline the statistical methods used, including survival analysis. Finally given the limited sample size, we now presented our manuscript in Current Oncology under the publication type Case Report to highlight the outcome in selected individual patient.
Below we address the reviewer for his/her constructive comments:
Reviewer #1 comment 1:
Lines 68 – 75 Report 1. The selection criteria of the cases (describe that it was about patients that did not respond to standard of care) 2. The inclusion and exclusion criteria in the two cohorts 3. If other cases were excluded why was that done? 4. Specify EPIC for readers not familiar with the USA health system
In lines 72-82, we have added detailed description of selection criteria and exclusions. We have added a brief explanation of EPIC in the Material and Methods, stating: "Epic is a healthcare software company based in the United States that provides electronic health system record (EHR) for managing patient records "
Reviewer #1 comment 2:
Lines 77 – 81 Specify abbreviations and cite in parenthesis the source of the staining reagents. Specify Tempus data in parenthesis.
We have provided manufacturer / antibody clone / catalogue number for Melan-A and SOX10 in lines 86-88.
We have added the abbreviations upon first use. These include (but not limited) the following abbreviations:
Ipilimumab (Ipi), temozolomide (TMZ), ipi 10mg/kg (Ipi10), Ipi 3mg/kg (Ipi3), RNA-sequencing (RNA-seq), central nervous system (CNS), Formalin-Fixed Paraffin-Embedded (FFPE), institutional review boards (IRB), polymerase chain reaction (PCR), Insertion-Deletion Variants (INDELs) and Single Nucleotide Variants (SNVs), Tumor Mutational Burden (TMB), Single Sample Gene Set Enrichment Analysis (ssGSEA), Gene Set Variation Analysis (GSVA), lactate dehydrogenase (LDH), magnetic resonance imaging (MRI), computed tomography (CT), positron emission tomography (PET), fluorodeoxyglucose (FDG), progressive disease (PD), partial response (PR), complete response (CR).
Additionally, we specified that sequencing was performed by Tempus Labs, Inc. (Chicago, IL).
Reviewer #1 comment 3:
Lines 83 – 106 Cite all sources when you report brand names
We have reviewed the manuscript and added manufacturer details for all reagents, including Melan-A and SOX10 (lines 86-88). In the original manuscript, lines 83 – 106 are description of WES and RNA-seq, mainly detailing the software tools used. We also included additional reference for the software, including reference for SIFT, PolyPhen and Hallmark pathway. In addition, we have now included the version of R packages that were utilized.
Reviewer #1 comment 4:
Line 107 The materials and methods part, lack a data analysis part explaining the statistical analysis performed.
First, we apologize for the unclear data analysis and lack of statistical analyses description. We have amended the Material and Methods section to include a “Statistical analysis“ section. Within this section we included details on ;
- Statistical test used to compare two groups (Between Ipi 10 a and Ipi 3)
- The use of Kaplan–Meier survival analysis for PFS and OS
- Software and package version used for analyses
- Definitions of statistical significance
Reviewer #1 comment 5:
Lines 108 – 212 It is not understandable whether these two case studies are included or excluded from the cohorts. In any case it is to be reconsidered how they should be presented. (if excluded must be included, if included they may be presented within each cohort presentation). Moreover, this part should be improved. The writing shall be more scientific- writing like than conference – presentation like.
These two was described /selected because the patients’ best response to IPI10 + TMZ was PR (partial response).
Reviewer #1 comment 6:
Lines 214 – 255 There is not evident what the authors mean reporting statistically significant or not significant. Shouldn’t they perform survival analysis? A Kaplan – Mayer chart must be presented also. These lines also must undergo extensive rewrite.
We apologize for the ambiguous description of the statistical significance. As suggested by the reviewer and addressed in comment #4, we clarified the definition of significant in the Statistical analysis section “A p-value of less than 0.05 was considered statistically significant”.
We thank the reviewer for the suggestion of the Kaplan-Mayer analysis. This result is now incorporated as Figure 4B in the manuscript. In line with our previous observation, patients treated with Ipi10+TMZ had longer PFS compared to patients treated with Ipi3+TMZ. We included these results in the manuscript.
In addition, the legend of Figure 4 now describes the statistical test employed and present the relevant statistical values.
Reviewer #1 comment 7:
The discussion also must be more precise and refer to data achieved. In my view the discussion should also be rewritten.
We discussed data presented in the result section. We’d appreciate more clarify what part of the discussion needs to be rewritten.
Reviewer #1 comment 8:
Line 364 Report if the consent was written.
Yes patients with molecular data analyzed and clinical photos presented provided written consents.
Reviewer 2 Report
Comments and Suggestions for Authors
In this manuscript, the authors provide evidence for a promising role of combination therapies with ipilimumab and temozolomide. Despite the small number of patients recruited, the study is of great importance and suitable for publication. Below you will find some comments/suggestions that can but do not have to be implemented.
1.) A general question regarding the level of immune cell infiltration in response to ipi+TMZ. Would it be possible to determine the immune score/ESTIMATE score of specimen? This would provide insights into the general level of infiltrated lymphocytes and is calculated in a GSEA-like fashion in R and based on a gene signature. For additional information please check: https://bioinformatics.mdanderson.org/estimate/rpackage.html
2.) Did you or will you investigate methyome landscapes of samples, considering that the role of MGMT and response to TMZ in glioblastoma.
3.) If possible (Don´t know whether this is needed/requested by the publisher) please make the RNAseq data available to the public e.g. via deposition in GEO.
4.) If you have the chance, perform immunohistochemistry on FFPE sections for CD3E or MHCs and CTLA4. It would be interesting to see (how is the expression level of CTLA4 and additional checkpoint molecules such as PD-1, PD-L1) whether protein levels are changed in response to therapies and depending on the location of tumors.
5.) Have you had the opportunity to investigate the brain metastases? can you estimate whether the BM emerged under therapy because they were intrinsically resistant to checkpoint inhibition or whether they develped earlier through dabrafenib/trametinib?
6.) The table shows a patient who was only 21 years old and died of nodular melanoma. Have you or will you also investigate germline mutations? In most cases, young patients suffering from such a severe disease carry mutations in BRCA2.
7.) Would be important to investiate the responses to ipi/TMZ in additional patient cohorts comprising more patients.
Author Response
Reviewer 2
In this manuscript, the authors provide evidence for a promising role of combination therapies with ipilimumab and temozolomide. Despite the small number of patients recruited, the study is of great importance and suitable for publication. Below you will find some comments/suggestions that can but do not have to be implemented.
We were pleased to read that the reviewer found our study “… of great importance and suitable for publication”. We also greatly appreciated that the reviewer provided suggestions to our manuscript.
Below we addressed the suggestions put forward by the reviewer.
Reviewer #2 comment 1:
A general question regarding the level of immune cell infiltration in response to ipi+TMZ. Would it be possible to determine the immune score/ESTIMATE score of specimen? This would provide insights into the general level of infiltrated lymphocytes and is calculated in a GSEA-like fashion in R and based on a gene signature. For additional information please check: https://bioinformatics.mdanderson.org/estimate/rpackage.html
We appreciate the reviewers insightful suggestion to conduct immune score analysis utilizing the ESTIMATE package. In response, we have performed ESIMATE analysis to conduct the immune score of our RNA-seq samples. The result is incorporated as part of Figure 2D. Our result indicates that a high “ImmuneScore” was present in sample M2, indicating significant immune cell inflitration. This finding aligns with the enrichment of interferon response observed from ssGSEA analysis and provides further insight into the patient response to high-dose ipilimumab + TMZ.
We have included these results in the Results section (“ The outcome of ESTIMATE analysis is a scoring matrix that infers the fraction of stromal cells (Stromal score) and immune cell fraction (Immune score) in individual samples. ESTIMATE analysis revealed that the liver metastatic (M2) has the highest Immune score. This observation was also supported by ssGSEA, where M2 showed an enrichment of immune-related gene sets including interferon alpha and gamma responses (Figure 2D) “) and in the Discussion section (“There are elevated Immune score indicating a robust immune presence within the tumor microenvironment, enriched gene expression and mutations of interferon alpha and gamma response pathways exclusively in the M2 sample, suggesting an enrichment of interferon response and representing a switch to a “hot tumor” after receiving low dose Ipi plus nivolumab treatment.“)
Reviewer #2 comment 2
Did you or will you investigate methyome landscapes of samples, considering that the role of MGMT and response to TMZ in glioblastoma.
We agree with the reviewer that Methyome is an important aspect that can correlate with response and worth looking into. We were not able to perform this in the current study but will definitely consider in the future.
Reviewer #2 comment 3
If possible (Don´t know whether this is needed/requested by the publisher) please make the RNAseq data available to the public e.g. via deposition in GEO.
We value the reviewer’s suggestion allow open access to the RNA-seq data. As such we deposited the data in a public repository - https://data.mendeley.com/datasets/nmb8csghjc/1. The repository contains the TPM, FPKM and expected count values provided by RSEM analysis. Upon acceptance of the manuscript, we will deposit analysis code to a GitHub repository for reproducibility.
Reviewer #2 comment 4
If you have the chance, perform immunohistochemistry on FFPE sections for CD3E or MHCs and CTLA4. It would be interesting to see (how is the expression level of CTLA4 and additional checkpoint molecules such as PD-1, PD-L1) whether protein levels are changed in response to therapies and depending on the location of tumors.
We agree with the reviewer this is an interesting and important topic. We are performing a separate study to correlate RNAseq immune scores with multiplex immune stains at protein level in a much larger cohort of patient samples.
Reviewer #2 comment 5
Have you had the opportunity to investigate the brain metastases? can you estimate whether the BM emerged under therapy because they were intrinsically resistant to checkpoint inhibition or whether they developed earlier through dabrafenib/trametinib?
This is an important question. We described the first case in page 4. This patient never received BRAF/MEK inhibitors (therefore they did not play a role in her brain progression). Her disease progressed to the brain and bones after three cycles of flipped dose ipi/nivo, but then responded to Ipi10/TMZ. As we discussed, ipi dosing is important, and for some patients (not all), they do need higher dose of IPI to activate the immune response / enhance the activation.
Reviewer #2 comment 6
The table shows a patient who was only 21 years old and died of nodular melanoma. Have you or will you also investigate germline mutations? In most cases, young patients suffering from such a severe disease carry mutations in BRCA2.
We thank the reviewer for highlighting the importance of including germline mutation analysis. At the time of treatment and data collection, germline genetic testing was not performed for this patient. We acknowledge the importance of investigating germline mutations in young patients with aggressive melanoma. In future studies, we plan to incorporate germline genetic analysis for such patients to enhance our understanding of the genetic factors contributing to early-onset melanoma.
Reviewer #2 comment 7
Would be important to investigate the responses to ipi/TMZ in additional patient cohorts comprising more patients.
We agree that studying additional patients would provide more robust data and strengthen the conclusion regarding the efficacy of high dose ipi + TMZ in advance melanoma. As our study is a retrospective analysis with a small sample size from a single institution, we acknowledge the limitations this imposes on the generalizability of our findings. We hope that our study will encourage further research and prospective clinical trials involving larger patient populations to validate these findings and explore the underlying mechanisms of the observed responses. In the revised manuscript, we have included a statement in the Discussion section acknowledging the need for larger studies and advocating for future research in this area.
“This study's small sample size and retrospective nature limit the generalizability of our findings. We recognize the importance of investigating the efficacy and safety of high-dose ipilimumab plus temozolomide in larger, more diverse patient cohorts. Future prospective studies with expanded patient populations are warranted to validate our results and further elucidate the potential role of this combination therapy in advanced melanoma treatment”
Reviewer 3 Report
Comments and Suggestions for Authors
The manuscript entitled: "Dramatic responses to high dose ipilimumab plus te- 2 mozolomide after progression on standard or low dose ipili- 3 mumab in advanced melanoma" raises an interesting scientific issue regarding the combination therapy of melanoma based on high dose ipilimumab plus te- 2 mozolomide. This is a retrospective review based on studies conducted at the Huntsman Cancer Institute from June 1, 2014 to June 30, 2022.
The manuscript requires major revision.
1. The manuscript presents descriptions of two clinical case reports with the BRAF V600E mutation. This should be noted in the title of the paper.
2. Another problem is the description of the selected group of patients treated with Ipi 10 mg/kg + TMZ (orange) 225 and Ipi 3 mg/kg + TMZ (blue) (Table I).
a. The selection of these patients is not clearly explained.
b. A ​​very important issue in the case of melanoma due to its heterogeneity is the mutation status in the BRAF, RAS, PTEN genes. The mutation status in these genes is responsible for, among other things, drug resistance and potential success of therapy.
c. The data should be supplemented and the problem of mutations should be discussed in relation to the treatment regimen and the results obtained.
Author Response
Reviewer 3
The manuscript entitled: "Dramatic responses to high dose ipilimumab plus te- 2 mozolomide after progression on standard or low dose ipili- 3 mumab in advanced melanoma" raises an interesting scientific issue regarding the combination therapy of melanoma based on high dose ipilimumab plus te- 2 mozolomide. This is a retrospective review based on studies conducted at the Huntsman Cancer Institute from June 1, 2014 to June 30, 2022.
The manuscript requires major revision.
We thank the reviewer for his/her time and were pleased to find that the reviewer finds our manuscript “rises an interesting scientific issue regarding the combination therapy of melanoma based on high dose…”.
Reviewer #3 comment 1
The manuscript presents descriptions of two clinical case reports with the BRAF V600E mutation. This should be noted in the title of the paper.
This study's small and we feel it is hard to make this conclusion based on two cases.
Reviewer #3 comment 2
Another problem is the description of the selected group of patients treated with Ipi 10 mg/kg + TMZ (orange) 225 and Ipi 3 mg/kg + TMZ (blue) (Table I).
- The selection of these patients is not clearly explained.
Please refer to the materials and methods section and response to reviewer 1 regarding the selection and exclusion criteria.
- A ​​very important issue in the case of melanoma due to its heterogeneity is the mutation status in the BRAF, RAS, PTEN genes. The mutation status in these genes is responsible for, among other things, drug resistance and potential success of therapy. The data should be supplemented and the problem of mutations should be discussed in relation to the treatment regimen and the results obtained.
We fully agree mutations status is important factors that can influence response and resistance to immunotherapies. We did exploratory analysis presented in Figure 2B (and raw data will be shared). This is too small of a study to make correlations with response but this important aspect should be further explored in future larger cohort of studies.
Round 2
Reviewer 1 Report
Comments and Suggestions for Authors
Dear Authors
Thank you for adressing my concerns about your manuscript.
Some minor ammendments shall be considered in the discussion part. When discussing results it is to note that we should refer to the results themselves. For example if we are speaking about the differences in survival between the two cohorts it is good to cite the relevant figure in parenthesis in order to help the reader spot the part of the results you are discussing in that special section. That means also that you should discuss first of all the two case studies then the cohorts discussing the figures and tables one by one (for example even in the charcteristics it is to discuss that females in the first cohort were 50% but in the second only the 1/6 or other special characteristics.)
Overall i belive that your manuscript merits to be considered for publication.
Author Response
Reviewer 1:
Dear Authors
Thank you for adressing my concerns about your manuscript.
Some minor ammendments shall be considered in the discussion part. When discussing results it is to note that we should refer to the results themselves. For example if we are speaking about the differences in survival between the two cohorts it is good to cite the relevant figure in parenthesis in order to help the reader spot the part of the results you are discussing in that special section. That means also that you should discuss first of all the two case studies then the cohorts discussing the figures and tables one by one (for example even in the charcteristics it is to discuss that females in the first cohort were 50% but in the second only the 1/6 or other special characteristics.)
Overall i belive that your manuscript merits to be considered for publication.
Response to the Reviewer: Thank you for these additional recommendations. Figures are now referenced in discussion. The difference in gender is added to results – cohort section.
Reviewer 3 Report
Comments and Suggestions for Authors
Unfortunately, the authors of the manuscript did not make the necessary changes.
The title of the manuscript misleads the reader into thinking that it is an extensive study, when in fact it concerns the description of two cases.
The authors did not address the remaining comments, including the discussion in the text of the status of changes in the BRAF gene.
I believe that in its current form the manuscript is not suitable for publication.
Author Response
Reviewer 3:
Comments and Suggestions for Authors
Unfortunately, the authors of the manuscript did not make the necessary changes.
The title of the manuscript misleads the reader into thinking that it is an extensive study, when in fact it concerns the description of two cases.
The authors did not address the remaining comments, including the discussion in the text of the status of changes in the BRAF gene.
I believe that in its current form the manuscript is not suitable for publication.
Response to the Reviewer: Thank you for these additional comments. We fully agree the importance to indicate both responders had BRAF mutant melanoma. Given this is a small cohort and another patient in the cohort who had initial mixed response had BRAF wild melanoma, we feel it may not be generalized to all BRAF mutant melanoma in the title. However, we did add this very relevant information in the abstract.
In lines 69-81, we have added detailed description of selection criteria and exclusions. Progression on standard of care was not required inclusion criteria; however all patients were heavily pre-treated with standard of care therapies as noted in the results and discussion.
Round 3
Reviewer 1 Report
Comments and Suggestions for Authors.